



# Effects of temporal averaging on short-term irradiance variability under mixed sky conditions

Gerald M Lohmann[1] and Adam H Monahan[2]

[1]Energy Meteorology Group, Institute of Physics, Oldenburg University, Germany
[2]School of Earth and Ocean Sciences, University of Victoria, Canada

*Correspondence to:* Gerald M Lohmann (gerald.lohmann@uol.de)

**Abstract.** Characterizations of short-term variability in solar radiation are required to successfully integrate large numbers of photovoltaic power systems into the electrical grid. Previous studies have used ground-based irradiance observations with a range of different temporal resolutions, and a systematic analysis of the effects of temporal averaging on the representa-

tion of variability is lacking. Using high-resolution surface irradiance data with original temporal resolutions between 0.01 s and 1 s from six different locations in the Northern Hemisphere, we characterize the changes in representation of temporal variability resulting from time averaging. In this analysis, we condition all data to states of mixed skies, which are the most potentially problematic in terms of local PV power volatility. Statistics of clear-sky index $k^*$ and its increments $\Delta k^*_\tau$ (i.e., normalized surface irradiance and changes therein over specified intervals of time) are considered separately. Our results indicate

that a temporal averaging time scale of around 1 s marks a transition in representing single-point irradiance variability, such that longer averages result in substantial underestimates of variability. Higher-resolution data increase the complexity of data management and quality control without appreciably improving the representation of variability. The results do not show any substantial discrepancies between locations or seasons.

## 1 Introduction

Both the installed capacity and the number of photovoltaic (PV) power systems are increasing in many regions of the world (Solar Power Europe (SPE), 2016). Unlike conventional power plants, PV systems are characterized by non-dispatchable and highly variable power production on spatiotemporal scales ranging from days to seconds and from hundreds of kilometers to meters (Perez et al., 2016). This instrinsic PV power variability complicates electrical grid operation and may negatively impact power quality and grid stability (Stetz et al., 2015). As changes in PV power are primarily determined by cloud-induced changes

in solar irradiance, a comprehensive data-driven characterization of irradiance variability can help mitigate the risks associated with the above-mentioned problems. While satellite-derived irradiance data provide large spatial coverage, their spatial and temporal resolutions are limited, so that high-resolution ground-based irradiance measurements are needed to analyze local short-term variability (Lohmann et al., 2016).

Previous PV-related studies on variability in ground-based solar radiation observations utilized data with a range of different

temporal resolutions. While early studies focused on hourly to daily timescales (e.g. Liu and Jordan, 1960), later analyses were



**Table 1.** Previous studies of short-term irradiance variability and the temporal resolutions considered.

| Temp. res. | Example references |
|---|---|
| 300 s | Jurado et al. (1995); Skartveit and Olseth (1992) |
| 60 s | Suehrcke and McCormick (1988); van Haaren et al. (2014) |
| 20 s | Hoff and Perez (2010); Perez et al. (2011, 2012) |
| 10 s | Hoff and Perez (2012); Munkhammar et al. (2017); Widén (2015) |
| 5 s | Woyte et al. (2006, 2007) |
| 1 s | Anvari et al. (2016); Bosch and Kleissl (2013); Calif et al. (2013); Dyreson et al. (2014); Hinkelman (2013); Lave et al. (2012); Lohmann et al. (2016, 2017); Lorenzo et al. (2015a); Madhavan et al. (2017); Marcos et al. (2011a, b); Monger et al. (2016); Perpiñán and Lorenzo (2011); Schmidt et al. (2016); Tabar et al. (2014) |
| 0.04 s | Torres Lobera et al. (2013) |
| 0.01 s | Gagné et al. (2016); Yordanov et al. (2013a, b) |

often geared towards finer temporal resolutions between 300 s and 0.01 s (Table 1). With regards to PV power applications, there is no consensus as to the proper temporal resolution of irradiance measurements needed to capture all relevant variability. The larger the panel-covered area of a PV system, the less variable in general is its power output compared to a single-point irradiance measurement (van Haaren et al., 2014), especially on short sub-minute timescales (Marcos et al., 2011a, b). Thus,

5     high temporal resolutions on the order of seconds and shorter may not be required to monitor large utility-scale PV plants (Woyte et al., 2013), while minute-averaged data may be too coarsely resolved (van Haaren et al., 2014). When considering smaller rooftop PV systems and/or partial shading (which can strongly reduce an inverter's power output as soon as a few connected modules are shaded; Belhaouas et al., 2017), previous research has shown that the temporal resolution needed to capture irradiance variability on all time scales may however be as small as 0.1 s (Torres Lobera et al., 2013; Yordanov et al.,

10     2013b) or 0.4 s (Gagné et al., 2016).

Those studies which argued the need for sub-second resolutions made this determination based on various different lines of reasoning, namely by

1. using the 2nd temporal derivative of irradiance as a measure for instantaneous variation, defining the daily minimum (negative) value of this derivative as each day's strongest instantaneous irradiance variation, and finally calculating a



hypothetical optimal averaging time based on an acceptable error of $10\,\mathrm{Wm}^{-2}$ and an assumed parabolic shape of the variation for each of a few hundred days in spring and summer (Yordanov et al., 2013b);

2. analyzing the sample standard deviation of irradiance as a function of averaging time measured during 7 hours on a single summer's day (Torres Lobera et al., 2013); and

3. separately studying two variability metrics as functions of averaging time for 7 selected days (Gagné et al., 2016).

In addition to these methodical differences and relatively short datasets, each study was limited to a single geographic area, namely Southern Norway (Yordanov et al., 2013b), Southern Finland (Torres Lobera et al., 2013), and Eastern Canada (Gagné et al., 2016).

    In this paper, we combine high-resolution irradiance data originating from six different locations in the Northern Hemi-
sphere in order to systematically assess the biases in representation of temporal irradiance variability resulting from temporally averaged measurements. The time series feature original temporal resolutions between 0.01 s and 1 s, and four of the sites have records of at least one full year, enabling seasonal comparisons of short-term averaging effects for the first time. We derive estimates of clear-sky index (i.e. surface irradiance normalized by its clear-sky value) and its increments (i.e. changes over fixed time intervals), and analyze the respective time series' standard deviations as functions of averaging time scale. In this
analysis, we condition all data to states of mixed skies, which are the most potentially problematic in terms of local PV power volatility. Details on all datasets and utilized methods are presented in Sec. 2, with the results and discussion following in Sec. 3, and conclusions in Sec. 4.

## 2   Material and methods

### 2.1   Available irradiance datasets

The global horizontal irradiance data used in this study were collected near

    1. Alderville, Ontario, Canada,

    2. Varennes, Québec, Canada,

    3. Jülich, Germany,

    4. Oldenburg, Germany,

5. Oahu, Hawaii, USA, and

    6. Tucson, Arizona, USA,

using different types of photodiode-based pyranometers with raw temporal resolutions $T_0$ ranging from 0.01 s through 1 s. Table 2 summarizes the different measurement sites' coordinates, available recording periods, number of pyranometers $N_{pyr}$, original temporal resolutions $T_0$, and the respective total durations of mixed-sky conditions available for the analyses.



**Table 2.** Details of the available clear-sky index data sets: location names and their respective latitudes and longitudes, the available recording periods, number of pyranometers $N_{pyr}$, original temporal resolutions $T_0$, and total durations of mixed-sky conditions available for the analyses.

| Location | Lat. [°] | Lon. [°] | Available period | $N_{pyr}$ | $T_0$ [s] | Mixed skies [h] |
|----------|----------|----------|------------------|-----------|-----------|-----------------|
| Alderville | 44.19 | -78.10 | 01/2015 – 03/2017 | 3 | 0.01 | 1113 – 1126 |
| Varennes | 45.62 | -73.39 | 10/2015 – 11/2016 | 2 | 0.01 | 581 – 595 |
| Jülich | 50.89 | 6.43 | 04/2013 – 07/2013 | 5 | 0.1 | 85 – 145 |
| Oldenburg | 53.15 | 8.17 | 05/2015 – 12/2016 | 3 | 0.1 | 1274 – 1304 |
| Oahu | 21.31 | -158.08 | 03/2010 – 11/2011 | 6 | 1.0 | 1628 – 1655 |
| Tucson | 32.07 | -110.84 | 04/2014 – 06/2014 | 5 | 1.0 | 83 – 99 |

The Canadian data from Alderville and Varennes were measured using LI-COR LI-200S pyranometers, with samples being taken every 0.001 s, and their averages evaluated every 0.01 s. Gagné et al. (2016) provide detailed information about the locations, data aquisition units and the local character of irradiance variability in Alderville and Varennes. We use a subset of three sensors from Alderville providing data from January 2015 through March 2017, and two sensors from Varennes, with

data being available from October 2015 through November 2016. The inter-sensor distances are on the order of 100 m each, and the two locations are about 400 km apart.

The data set collected near Jülich consists of a five-sensor subset of a larger pyranometer network, which was deployed during the HD(CP)$^2$ Observational Prototype Experiment (HOPE; Macke et al., 2017) between 2 April and 24 July 2013, using as many as 99 EKO ML-020VM sensors. The selected sensors are evenly distributed accross the entire domain and separated by

a few kilometers each. Madhavan et al. (2016) detail the HOPE campaign with respect to this pyranometer network and provide further information about data acquisition and quality control. Several studies have already used 1 s averages derived from the original 0.1 s irradiance samples to characterize the averages' spatiotemporal variability (Lohmann et al., 2016; Madhavan et al., 2017), and to evaluate sky-imager-based irradiance retrievals (Schmidt et al., 2016). For our present analyses of the five-sensor subset, we use the original 0.1 s irradiance samples.

In Oldenburg, three EKO ML-01 pyranometers have been continually recording 0.1 s irradiance samples since 28 April 2015 as part of an in-house weather observation system of the university's Energy Meteorology Group. The sensors are horizontally mounted on a university building rooftop with inter-sensor distances of about 15 m, and subject to bi-weekly maintenance, including verification of the horizontal orientation and cleaning of the glass dome. The quality of the measurements was verified for randomly selected periods using additional measurements from a co-located well-established thermopile pyranometer

(Schmidt, 2017, personal communication). We use data from 1 May 2015 through 31 December 2016 in this paper.

Near Kalaeloa Airport on Oahu, the National Renewable Energy Laboratory (NREL) performed a measurement campaign using 17 LI-COR LI-200 pyranometers from March 2010 though November 2011, collecting irradiance data every 1 s (Sen-





gupta and Andreas, 2010). These data have previously been used and analyzed in several irradiance variability studies (e.g. Anvari et al., 2016; Aryaputera et al., 2015; Hinkelman, 2013; Madanchi et al., 2017; Munkhammar et al., 2017). In the present paper, we use data from the six-pyranometer subset located on the premises of Kalaeloa Airport, each separated by a few 100 m.

The data we use from Tucson were measured between 5 April and 30 June 2014 with a temporal resolution of 1 s. While
the corresponding measurement campaign featured different photodiode-based sensors, we limit ourselves to a subset of five Apogee SP-212 pyranometers, so that all Tucson data used in this paper originate from the same type of sensor. The inter-sensor distances of the subset are on the order of 100 m. Lorenzo et al. (2015a) have previously documented details of the entire data set, and used it to evaluate solar irradiance forecasts.

## 2.2 Data preprocessing

All available irradiance time series feature infrequent drop-out periods, during which the measured irradiance is suddenly reduced to almost $0\,\mathrm{Wm}^{-2}$ and then resumes a reasonable value after a short while (seconds to minutes). These unreasonably low readings can be in part associated with times of regular maintenance, during which a sensor's glass dome is covered with a piece of cloth. Additionally, they can be caused by e.g. birds, insects, or leaves temporarily occupying the small sensor area. To eliminate these drop-out periods from the data, we remove a time window of $\pm 60\,\mathrm{s}$ around each irradiance value falling below
a fixed threshold, taken to be $25\,\mathrm{Wm}^{-2}$.

Next, we normalize all irradiance measurements $G$ to their respective clear-sky values $G_{clear}$, which we calculate using the clear-sky model described by Fontoynont et al. (1998), to obtain time series of clear-sky index

$$k^* = \frac{G}{G_{clear}}. \tag{1}$$

While irradiance is subject to astronomically-determined variations and trends, the clear-sky index is convenient for comparing
short-term variability at different locations, and during different times of the day or year. However, $k^*$ estimates are highly uncertain for low solar elevation angles (Lave et al., 2012), and we thusly remove all data associated with elevation angles below 15° at this point.

In order to extract mixed sky periods from the clear-sky index time series, we employ a simple classification scheme based on dividing all time series into non-overlapping 900 s blocks, and calculating the individual blocks' clear-sky index standard
deviations (Lohmann et al., 2016). If the standard deviation of $k^*$ exceeds a fixed threshold (taken to be 0.18, based on the results of Lohmann et al., 2016), we classify the corresponding 900 s block as mixed, and retain it for our analyses.

The resulting ranges of durations of mixed sky conditions are quoted in Table 2, and strongly differ between locations. While Oahu, Oldenburg, and Alderville each provide a total of well over 1000 h of mixed-sky data, the brief campaigns around Jülich and Tucson feature mixed-sky records that are about an order of magnitude shorter. From Varennes, an intermediate number of
more than 500 h of mixed-sky observations are available.




## 2.3 Characterizing variability

Using all data classified as mixed-sky conditions from each pyranometer of each location, we compute time series of $k^*$ averages using a range of different averaging time scales $0.01\,\mathrm{s} < T < 900\,\mathrm{s}$. In this calculation, we apply non-overlapping moving windows of length $T$ to each $900\,\mathrm{s}$ block and calculate the mean clear-sky index within each window. If fewer than

$95\,\%$ of the highest-resolution data within a period $T$ are available, the resulting average is classified as missing. Otherwise the missing values in the raw data are simply neglected in the computation. We found this threshold of $95\,\%$ to be a good compromise, minimizing the number of missing data while maintaining the robustness of the mean estimates. With appreciably higher thresholds, a small number of missing high-resolution data points could cause unreasonably many long-term averages to be missing; while for considerably lower thresholds, there would be too much variation in the number of data points entering

the computation of means of nominal averaging time $T$.

Figure 1 presents two illustrative examples of $k^*$ time series under mixed sky conditions using temporal averaging scales across up to six orders of magnitude. The $1000\,\mathrm{s}$ period and its $100\,\mathrm{s}$ excerpt shown in panels (a,b) are typical examples of high temporal clear-sky index variability, while the $100\,\mathrm{s}$ period and its $10\,\mathrm{s}$ excerpt shown in panel (c,d) were specifically selected as a case featuring an unusually volatile time series with short-term variability on all time scales. In all panels, an evident

decrease of temporal variability can be observed for increasing averaging periods $T$, with the most pronounced reductions starting at the transition from $T = 1\,\mathrm{s}$ to $T = 10\,\mathrm{s}$. The variance reduction for averaging periods $T < 1\,\mathrm{s}$ is relatively large for the time series shown in (d).

To characterize clear-sky index variability as a function of averaging time scale, we will first compare probability distributions of $k^*$ at each of the available sensors at all locations for averaging times $T = 1\,\mathrm{s}$ and $T = 100\,\mathrm{s}$. For each sensor at each

location, we then estimate the sample standard deviation

$$\sigma_T^{k^*} = \sqrt{\frac{1}{N_T - 1} \sum_{t=1}^{N_T} (k_T^*(t) - \overline{k_T^*})^2} \qquad (2)$$

for averaging periods of $0.01\,\mathrm{s} < T < 900\,\mathrm{s}$, where $N_T$ is the total number of available data points for averages of length $T$, and $\overline{k_T^*}$ denotes the corresponding arithmetic mean value of $k^*$. In order to facilitate comparisons between locations, we also use a normalized clear-sky index standard deviation

$$\hat{\sigma}_T^{k^*} = \frac{\sigma_T^{k^*}}{\sigma_{T=1\,\mathrm{s}}^{k^*}} \qquad (3)$$

for each time scale and location, using the standard deviation associated with $T = 1\,\mathrm{s}$ as a normalization factor. The decline of $\hat{\sigma}_T^{k^*}$ as a function of $T$ quantifies the reduction of clear-sky index variability for increasing averaging time periods. Finally, we repeat the calculations of Eqn. 2 and 3 for each season using the longer data from Alderville, Varennes, Oldenburg, and Oahu. In this seasonal analyses, winter is defined as December through February, spring as March through May, summer as

June through August, and autumn as September through November. In the normalization factor in Eq. 3, we always use the full-year standard deviations $\sigma_{T=1\,\mathrm{s}}^{k^*}$ derived without conditioning by season.





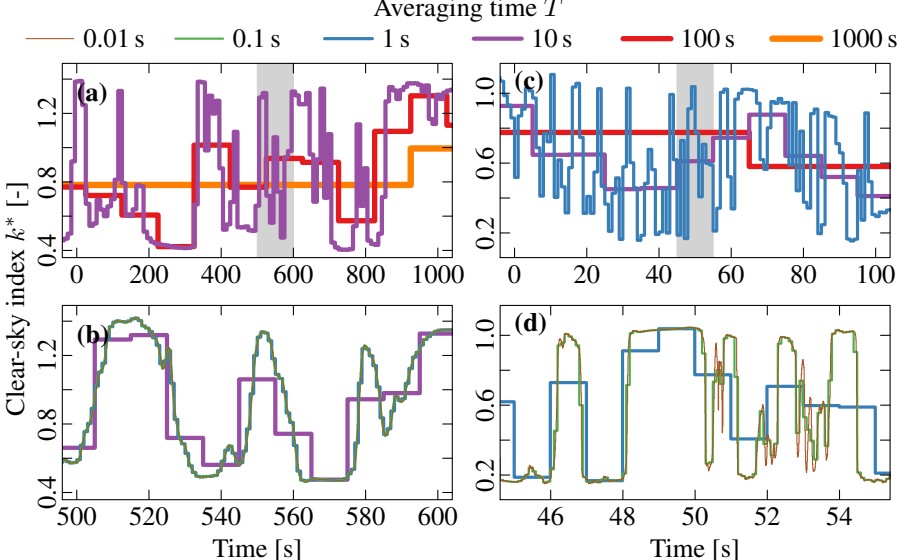

**Figure 1.** Two examples of highly-variable clear-sky index $k^*$ time series under mixed sky conditions: panels (a,b) show a typical example of a 1000 s period using six averaging times $T$, namely (a) $T = 1000\,\text{s}$, $T = 100\,\text{s}$, and $T = 10\,\text{s}$, and (b) $T = 10\,\text{s}$, $T = 1\,\text{s}$, $T = 0.1\,\text{s}$, and $T = 0.01\,\text{s}$; panels (c,d) show an unusual 100 s example with evident variability down to the shortest time scales with averaging periods (a) $T = 100\,\text{s}$, $T = 10\,\text{s}$, and $T = 1\,\text{s}$, and (b) $T = 1\,\text{s}$, $T = 0.1\,\text{s}$, and $T = 0.01\,\text{s}$. The time series were measured in Varennes on March 2, 2016, starting at 15:01:35 EDT (a,b), and on September 21, 2016, starting at 15:06:35 EDT (c,d). The gray areas in the top row indicate the excerpts presented in the bottom row.

The standard deviation of $k^*$ is independent of the observations' ordering in time, and thus does not quantify how quickly the values can change. In order to consider the effect of temporal averaging on the estimates of rates of change, we compute $k^*$ increments

$$\Delta k^*_\tau(t) = k^*(t + \tau) - k^*(t) \tag{4}$$

for different time lags $\tau$ on the distinct averaging time scales $T = 0.01\,\text{s}$, $T = 0.1\,\text{s}$, $T = 1\,\text{s}$, $T = 10\,\text{s}$, and $T = 100\,\text{s}$. Similar to the analyses of $k^*$ variability, we first explore distributions of $\Delta k^*_\tau$ for $T = 100\,\text{s}$ and $T = 1\,\text{s}$, using $\tau = 100\,\text{s}$ and $\tau = 1\,\text{s}$. Then we calculate clear-sky index increment standard deviations $\sigma_T^{\Delta k^*_\tau}$ for the same five averaging times and a range of $0.01\,\text{s} < \tau < 900\,\text{s}$ analogous to Eq. 2, and derive their normalized version $\hat{\sigma}_T^{\Delta k^*_\tau}$ using $T = 1\,\text{s}$ (as in Eq. 3). As before, we finally estimate $\sigma_T^{\Delta k^*_\tau}$ and $\hat{\sigma}_T^{\Delta k^*_\tau}$ conditioned to the four seasons using the year-round observations from Alderville, Varennes, Oldenburg, and Oahu.





## 3   Results and discussion

### 3.1   Clear-sky index variability

For the two averaging times $T = 1\,\mathrm{s}$ and $T = 100\,\mathrm{s}$, Fig. 2 presents the probability density functions of clear-sky index, based on all available mixed-sky periods from the six locations. The 1 s averages feature pronounced bimodal distributions for all

locations. The two modes are located slightly above $k^* \simeq 1$ (corresponding to full sunlight exposure), and within a range of $0.3 \lesssim k^* \lesssim 0.5$ (corresponding to cloud shadow coverage), with modest differences between locations. The probability minimum between the peaks is less pronounced at Tucson as at other locations. This distinct bimodality is substantially weaker for the distributions of 100 s averages. Instead, while the high-value peaks are still very pronounced (but located at somewhat smaller values compared to the 1 s averages), the low-value peaks are replaced by flat shoulders in the distribution ranging

from $k^* \simeq 0.4$ through $k^* \simeq 0.9$.

We interpret the minor differences in the distinct $k^*$ values at which the peaks occur as results of imperfect simulations of clear-sky irradiance, and hence a biased estimation of clear-sky index as per Eq. 1, as well as the fact that each location uses distinct types of pyranometers with potentially varying calibration performances and possibly marginally tilted orientations. The very high values $k^* > 1$ being frequently recorded at all locations are caused by short-term cloud enhancement (see e.g.

Schade et al., 2007), while the comparatively high spread and rough structures of Jülich and Tucson (especially for $T = 100\,\mathrm{s}$) in panels (c) and (f) are due to sampling variability resulting from the relatively short periods of available observations (cf. Table 2).

The normalized clear-sky index standard deviation $\hat{\sigma}_T^{k^*}$ decays as a function of averaging time $T$ (Fig. 3). The values remain close to $\hat{\sigma}_T^{k^*} \simeq 1$ for values $T \lesssim 5\,\mathrm{s}$, and then begin to rapidly decrease with increasing $T$. For example, around $T \simeq 50\,\mathrm{s}$, the

normalized standard deviation has dropped to $90\,\% \lesssim \hat{\sigma}_T^{k^*} \lesssim 95\,\%$, and around $T \simeq 500\,\mathrm{s}$, it has become as low as $60\,\% \lesssim \hat{\sigma}_T^{k^*} \lesssim 67\,\%$, depending on the location. For averaging times $T < 1\,\mathrm{s}$, the normalized standard deviations do not go appreciably above $\hat{\sigma}_T^{k^*} \simeq 1$ (the highest value is $\hat{\sigma}_T^{k^*} = 1.0005$ for $T = 0.01\,\mathrm{s}$; not shown).

In general, the structures of $\hat{\sigma}_T^{k^*}$ at the different locations and their corresponding normalization factors $\sigma_{T=1\,\mathrm{s}}^{k^*}$ compare well, with relatively small differences in the rate at which variability declines and in the absolute values of $\sigma_{T=1\,\mathrm{s}}^{k^*}$. No direct

relation is evident between the order in which the normalized standard deviation structures decline, and the order of absolute standard deviations associated with $T = 1\,\mathrm{s}$. For example, Oldenburg and Oahu feature almost identical values of $\sigma_{T=1\,\mathrm{s}}^{k^*}$ (0.37 and 0.36, respectively), but correspond to end members in terms of variability decay rates, with Oahu featuring the fastest and Oldenburg the slowest decline. Likewise, the normalized structures of Oldenburg and Jülich both decline with a similar rate, but their values of $\sigma_{T=1\,\mathrm{s}}^{k^*}$ differ by approximately 10 % (0.37 and 0.33, respectively). As discussed above, imperfectly simulated

clear-sky irradiance and possible differences in pyranometer calibration may contribute to the differences in absolute standard deviations for relatively close locations (like Oldenburg and Jülich). Differences between locations are largely removed by normalization. Even though the cloud climatologies are different at the different locations (for example, the Oahu data are strongly influenced by passing trade wind cumulus clouds, while the mid-latitude locations experience a more diverse variety of different cloud and weather phenomena), the statistical structures are very similar across locations.





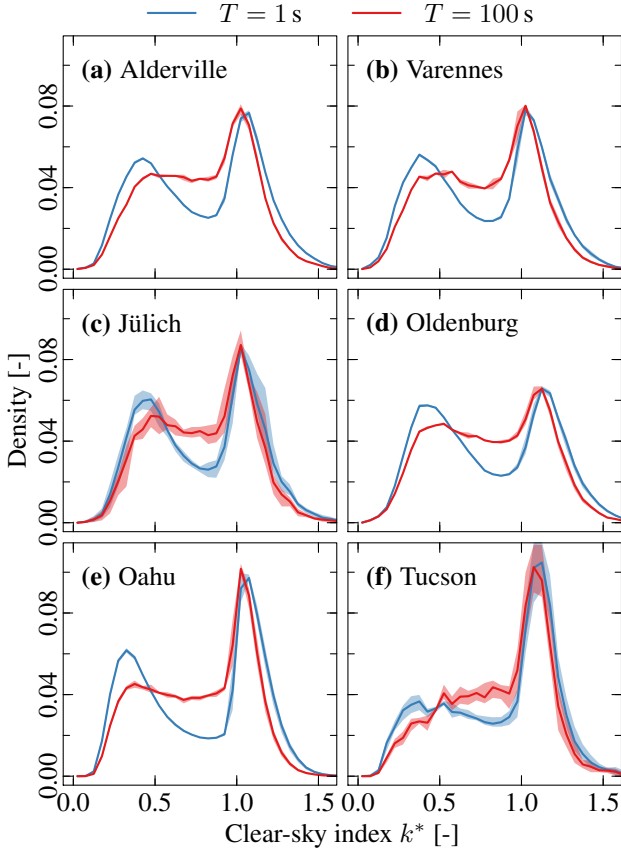

**Figure 2.** Distributions of clear-sky index $k^*$ under mixed-sky conditions for two averaging times $T = 1\,\mathrm{s}$ and $T = 100\,\mathrm{s}$ and all six lo-cations, estimated from histograms using a bin width of 0.05. The spread across multiple single-sensor distributions is indicated using semi-transparent coloring, while the solid lines are derived by averaging the single-sensor densities in each bin.

Using data from the four locations with measurements in all seasons, Fig. 4 shows $\hat{\sigma}_T^{k^*}$ as a function of averaging time separately for winter, spring, summer, and autumn. The respective durations of available mixed-sky data are additionally quoted in each panel. In general, the structures are very similar to the ones derived without conditioning by season (cf. Fig. 3), with the exception of winter periods in Alderville, Varennes, and Oahu showing a somewhat slower decrease of $\hat{\sigma}_T^{k^*}$ compared to the other seasons. Moreover, $\hat{\sigma}_T^{k^*}$ values associated with Alderville's autumn, as well as with Oldenburg's winter and spring are slightly higher than $\hat{\sigma}_T^{k^*}$ derived from all data (i.e. $\hat{\sigma}_{T=1\,\mathrm{s}}^{k^*} > 1$), while Oldenburg's summer exhibits lower values (i.e. $\hat{\sigma}_{T=1\,\mathrm{s}}^{k^*} < 1$). No systematic pattern of seasonal variations in $\hat{\sigma}_T^{k^*}$ exists among the different locations. These results imply that the various cloud types prevailing during the different seasons exert relatively little influence on the temporal averaging effects of irradiance variability under mixed-sky conditions.





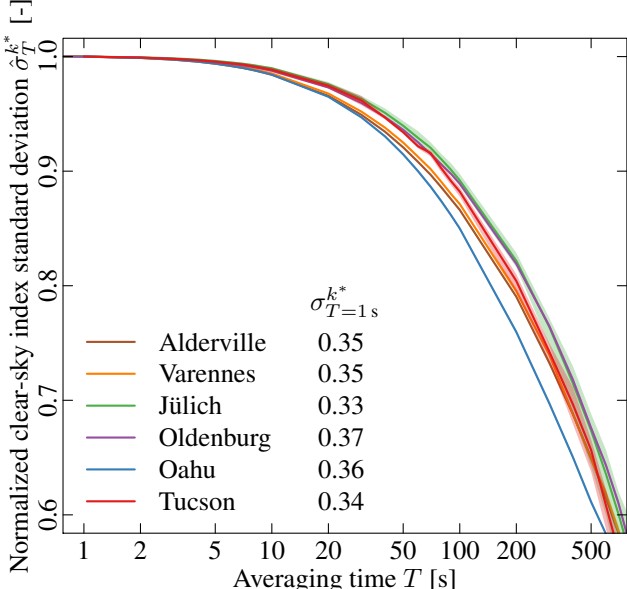

**Figure 3.** Structures of normalized clear-sky index standard deviation $\hat{\sigma}_T^{k^*}$ as a function of averaging time $T$ under mixed-sky conditions for all six available locations. The normalization factors $\sigma_{T=1\,\text{s}}^{k^*}$ are quoted for each location. The spread across multiple individual single-sensor structures is indicated using semi-transparent coloring, while the solid lines are derived by averaging the single-sensor structures for each $T$.

### 3.2 Increment variability

For three combinations of averaging time scales and increment time steps ($T = 1\,\text{s}$ and $\tau = 1\,\text{s}$; $T = 1\,\text{s}$ and $\tau = 100\,\text{s}$; $T = 100\,\text{s}$ and $\tau = 100\,\text{s}$), Fig. 5 shows distributions of clear-sky index increments, using all available mixed-sky periods from the six locations. At all locations, the resulting distributions exhibit global maxima at $\Delta k_\tau^* = 0$, with characteristic shapes. For $T = 1\,\text{s}$

5   and $\tau = 1\,\text{s}$, the distributions are chevron-shaped, with densities decreasing rapidly for increasing positive and decreasing negative increment values. For the same averaging time $T = 1\,\text{s}$ but the larger increment $\tau = 100\,\text{s}$, the distributions have broad shoulders, with two local maxima around $\Delta k_\tau^* \approx \pm 1$. Beyond these secondary maxima the tails decrease rapidly. For the longer averaging time $T = 100\,\text{s}$ and a corresponding increment time step of $\tau = 100\,\text{s}$, the distributions are more rounded around the distinct central peak.

10   With respect to the increment distributions of 1 s averages, an increase of the increment time step from $\tau = 1\,\text{s}$ to $\tau = 100\,\text{s}$ leads to an increase of high-magnitude increment probabilities by many orders of magnitude. While $\tau = 1\,\text{s}$ is too short an increment time step for strong variations between clear and cloudy states to occur frequently, a time step of $\tau = 100\,\text{s}$ frequently covers these transitions. Compared to $T = 1\,\text{s}$, averaging the time series using $T = 100\,\text{s}$ leads to a considerable underestimation of both magnitude (i.e. the distributions are narrower) and probabilities (i.e. the distributions' shoulders are less pronounced)

15   of strong 100 s increments.





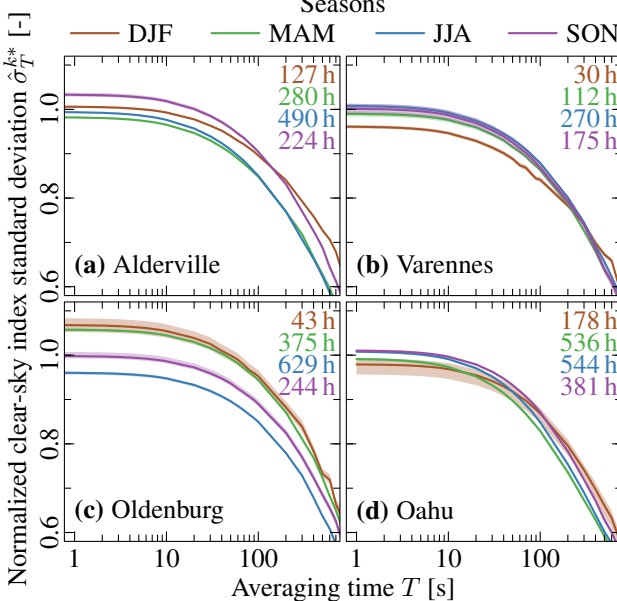

**Figure 4.** As in Fig. 3, for structures separately estimated for winter (December, January, February; DJF), spring (March, April, May; MAM), summer (June, July, August; JJA), and autumn (September, October, November; SON). The respective durations of available mixed-sky data are quoted in the appropriate color for each season and location. The normalization factors are the location-specific standard deviations $\sigma_{T=1\,\mathrm{s}}^{k^*}$ derived without conditioning by season (Fig. 3). The spread across individual single-sensor structures is indicated using semi-transparent coloring, while the solid lines are derived by averaging the single-sensor structures for each $T$.

While the increment distributions are strongly affected by averaging time and increment size, they are remarkably similar at all locations considered. The relatively small differences between locations (e.g., the exact values of the two secondary maxima for $T = 1\,\mathrm{s}$ and $\tau = 100\,\mathrm{s}$) may result from imperfections in the clear sky model, as well as sensor calibration and leveling, as discussed earlier. The different lengths of the distributions' tails are directly determined by the record lengths (short at Jülich

5  and Tucson; longer at the other locations). The estimates of the probability density tails are less robust than estimates within the core of the distribution, because they are typically based on a comparatively limited number of observations. In Varennes, for example, only about 100 occurences of the absolute increment value $|\Delta k_\tau^*| > 0.6$ are registered among the total of more than $2 \cdot 10^6$ data points for $T = \tau = 1\,\mathrm{s}$.

Comparing the distributions of clear-sky index for $T = 1\,\mathrm{s}$ (Fig. 2) with those of clear-sky index increments for $T = 1\,\mathrm{s}$ and

10  $\tau = 100\,\mathrm{s}$ (Fig. 5) allows further characterization of transitions from full sunlight exposure to full cloud shadow coverage (and vice versa). From the separation of the two distinct modes of the $k^*$ distributions, a clear-sky index difference of approximately 0.7 can be inferred between cloud-free and cloud-covered states. However, the secondary peaks of the increment distributions shown in Fig. 5 indicate transitions of $\pm 1$ in the clear-sky index to occur more frequently than changes of $\pm 0.7$. This disparity may be accounted for by considering short-term cloud enhancement: instead of a typical transition going directly from e.g.





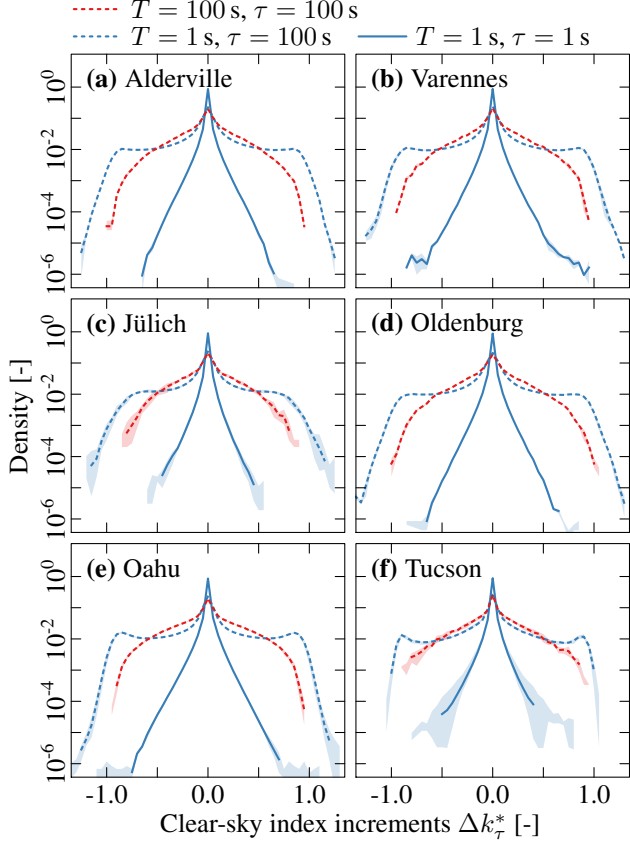

**Figure 5.** Location-specific distributions of clear-sky index increments $\Delta k_\tau^*$ under mixed-sky conditions for averaging times $T = 1\,\text{s}$ and $T = 100\,\text{s}$, and increment time steps $\tau = 100\,\text{s}$ and $\tau = 1\,\text{s}$, using a bin width of 0.05. The spread across individual single-sensor distributions is indicated using semi-transparent coloring for all bins with data from at least two sensors. The averages of the single-sensor densities are presented as solid or dashed lines for the bins that contain data from every sensor.

clear ($k^* \approx 1$) to cloudy ($k^* \approx 0.3$), cloud enhancement immediately precedes (or follows) cloud cover, so typical transitions occur between a cloud-enhanced state ($k^* \gg 1$) and a cloud-covered state ($k^* \approx 0.3$).

In order to further investigate the effect of increasing averaging times on increment variability, Fig. 6 presents both absolute and normalized clear-sky index increment standard deviations as a function of increment time step. The structures of $\sigma_T^{\Delta k_\tau^*}$

5 (left column) are effectively indistinguishable for $T = 0.01\,\text{s}$, $T = 0.1\,\text{s}$, and $T = 1\,\text{s}$, while structures associated with longer averaging periods of $T = 10\,\text{s}$ and $T = 100\,\text{s}$ deviate notably from their high-resolution counterparts. Note that because of differences in the resolution of the data at different locations, shorter averaging timescales can be considered at some stations (e.g. Varennes and Alderville) than at others (e.g. Oahu and Tucson). The display of normalized standard deviation $\hat{\sigma}_T^{\Delta k_\tau^*}$ (Fig. 6, right column) emphasizes the structures' differences, especially for small increment time steps. For any given averaging period

10 $T$, smaller increment time steps $\tau$ are associated with larger deviations of $\hat{\sigma}_T^{\Delta k_\tau^*}$ from unity (smaller increment steps are more





affected by averaging than larger ones). Similarly, for any given $\tau$, shorter averaging times $T$ coincide with values of $\hat{\sigma}_T^{\Delta k_\tau^*}$ closer to one (for a given increment step, longer averaging intervals have a bigger effect on variability).

Except for the anomalously low values of Varennes-based $\hat{\sigma}_T^{\Delta k_\tau^*}$ for $T = 0.1\,\mathrm{s}$ and $\tau \lesssim 0.5\,\mathrm{s}$ in panel (d), which we further discuss in Sec. 3.3, the structures of both absolute and normalized clear-sky index increment standard deviations resemble each

other well for all locations. Averages associated with $T \leq 1\,\mathrm{s}$ typically result in normalized increment standard deviations close to unity ($0.95 \leq \hat{\sigma}_T^{\Delta k_\tau^*} < 1$ for $\tau < 10\,\mathrm{s}$ and $\hat{\sigma}_T^{\Delta k_\tau^*} \approx 1$ for $\tau \gtrsim 10\,\mathrm{s}$), while averages of $T = 10\,\mathrm{s}$ and $T = 100\,\mathrm{s}$ systematically underestimate increment variability for all time steps. This result is a comprehensive quantification of the qualitative features seen in the example time series in Fig. 1. Imperfections in pyranometer configuration and clear sky irradiance estimation do not appreciably influence the normalized structures, and we interpret the minor differences between location-specific increment

standard deviations of 100 s averages as resulting from differences in local cloud conditions. For example, Oahu's typical trade wind cumulus clouds can be expected to generate relatively high variability on small time scales, and consequently, Oahu features smaller values of $\hat{\sigma}_T^{\Delta k_\tau^*}$ for $T = 100\,\mathrm{s}$ than the other locations.

For those locations with year-round measurements, Fig. 7 displays $\sigma_{T_0}^{\Delta k_\tau^*}$ and $\hat{\sigma}_T^{\Delta k_\tau^*}$ as functions of step size $\tau$ for different averaging timescales separately for winter, spring, summer, and autumn. The structures resemble those derived without condi-

tioning by season (cf. Fig. 6), with some quantitative differences. For example, wintertime values of $\sigma_{T_0}^{\Delta k_\tau^*}$ are slightly lower compared to the results derived from all data for Alderville, Varennes, and Oahu in panels (a, c, g), while $\hat{\sigma}_T^{\Delta k_\tau^*}$ is slightly higher for Oahu and $T = 100\,\mathrm{s}$ in panel (h). These winter-specific differences in increment variability structures do not differ by more than $\pm 10\,\%$ relative to the other seasons, and we consider them comparatively minor.

### 3.3  Peculiarity in Varennes data

Both in Fig. 6(d) and in Fig. 7(d), the normalized standard deviations of clear-sky index increments at Varennes indicate a strong reduction of variability for increment time steps of $\tau = 0.1\,\mathrm{s}$ when averaging the original high-resolution data ($T_0 = 0.01\,\mathrm{s}$) with an averaging time period of $T = 0.1\,\mathrm{s}$. Except for the winter data in Fig. 7(d), the increment standard deviation is reduced to about 90 % of its original value, which is considerably lower than the value associated with $T = 1\,\mathrm{s}$ and $\tau = 1\,\mathrm{s}$. This behavior matches neither the results obtained from Alderville for the same time scales (cf. Fig. 6(b) and Fig. 7(b)), nor the general

characteristics of the value of $\hat{\sigma}_T^{\Delta k_\tau^*}$ associated with $\tau = T$ to decrease with increasing $T$.

In order to investigate the issue, we separately derive $\hat{\sigma}_T^{\Delta k_\tau^*}$ for each individual 900 s mixed-sky period in Varennes using $T = 0.01\,\mathrm{s}$ and $T = 0.1\,\mathrm{s}$, as well as $\tau = 0.1\,\mathrm{s}$. The distribution of the resulting set of single-block normalized increment standard deviations is shown in panel (a) of Fig. 8. For the majority of mixed-sky blocks in Varennes, the values of normalized standard deviations are very close to unity and therefore comparable to those at other stations on the scales in question. For example, the

rightmost bin ($0.985 < \hat{\sigma}_T^{\Delta k_\tau^*} \leq 1.0$) alone contains 79 % of all data (note the logarithmic scaling). This result implies that the relatively low value of $\hat{\sigma}_T^{\Delta k_\tau^*}$ computed over all blocks is being influenced strongly by a very small number of peculiar periods.

An example time series of $k^*$ and $\Delta k_\tau^*$ associated with a very small value of $\hat{\sigma}_T^{\Delta k_\tau^*} = 0.57$ is shown in panels (b) and (c) of Fig. 8. This time series demonstrates that even within such a peculiar block, only a very small number of data points are responsible for the small block-specific normalized increment standard deviation. While the clear-sky index varies considerably



throughout the block, there are only 4 events for which the averaged time series of $T = 0.1\,\mathrm{s}$ differs evidently from the original measurements of $T = 0.01\,\mathrm{s}$. Within a few hundredths of a second, the original $k^*$ readings drop considerably and immediately return to their previous values during these 4 events. Averaging the time series over periods of $T = 0.1\,\mathrm{s}$ removes these events almost entirely.

A few such very rapid and short-lived changes of considerable magnitude on time scales on the order of the original temporal resolution (here: $T_0 = 0.01\,\mathrm{s}$), result in a relatively high increment standard deviation for small $\tau$. If each single one of these changes in the time series consists of a sufficiently small number of data points in a row, an averging time period $T_1$ of an order of magnitude higher than $T_0$ (here: $T_1 = 0.1\,\mathrm{s}$) can cause the short excursions to be averaged out almost completely, resulting in a much smaller increment standard deviation for small $\tau$ compared to the original temporal resolution, and hence a low value

of normalized increment standard deviation for $\tau \simeq T_1$.

     We have not established a clear cause of these extremely short reductions in irradiance. The observed short-term peculiarities are possibly a result of some kind of small objects briefly passing the sensor, for example birds, insects, or leaves. Although the same sensor type is used in Alderville and Varennes with the same original temporal resolutions, the Alderville data do not exhibit these rapid changes in $k^*$ resulting in particularly high values of $\hat{\sigma}_T^{\Delta k_\tau^*}$. A reason for this difference may be the

measuring setup and its surroundings: in Varennes, the pyranometers were mounted on tripods and scattered on grassland, with a considerable number of nearby trees, while in Alderville, the sensors were located within a multi-megawatt PV plant (Gagné et al., 2016). The absence of small values of $\hat{\sigma}_T^{\Delta k_\tau^*}$ in winter at Varennes (cf. Fig. 7) may also indicate that the peculiar events are caused by insects. However, the short record of only $30\,\mathrm{h}$ worth of mixed-sky data from Varennes in winter does not suffice for a conclusive diagnosis of the cause of these intermittent rapid changes.

**4    Conclusions**

Based on a unique set of irradiance measurements with original temporal resolutions between $0.01\,\mathrm{s}$ and $1\,\mathrm{s}$ from six locations in the Northern Hemisphere, we have characterized the effects of temporal averaging on short-term irradiance variability under mixed-sky conditions, for the entire year as well as for individual seasons. For this analysis, we have analyzed probability distributions of the clear-sky index and its short-term increments on a range of temporal scales between $0.01\,\mathrm{s}$ and $900\,\mathrm{s}$, and

studied each particular time series standard deviation as a function of averaging time scale and increment time step. These quantities were found to be largely independent of season and location for the data considered. The following main results thus apply to all of the available mixed-sky data we considered:

- Clear-sky index distributions are strongly bimodal on averaging timescales between $0.01\,\mathrm{s}$ and $1\,\mathrm{s}$. The peaks are separated by a clear-sky index difference of approximately 0.7 and respecively represent cloud-covered and cloud-free states.

- Clear-sky index increment distributions of $1\,\mathrm{s}$ averages and $100\,\mathrm{s}$ increment time steps indicate that instead of typical transitions going directly from e.g. clear ($k^* \approx 1$) to cloudy ($k^* \approx 0.3$), they occur from a cloud-enhanced state ($k^* \gg 1$) to a cloud-covered state ($k^* \approx 0.3$), and vice versa.



- For the clear-sky index, an averaging time $T \lesssim 5\,\text{s}$ is sufficient to capture all $k^*$ variability, while for averaging times $T$ beyond about $10\,\text{s}$ cause the suppression of small-scale variability increases rapidly (e.g., one-minute averages only retain about $90\,\%$ of the original $k^*$ standard deviation).

- For clear-sky index increments, a small averaging time of $T = 1\,\text{s}$ reduces increment standard deviation by approximately $5\,\%$ on the same scale of $\tau = 1\,\text{s}$, but effectively retains all variability information for increment time steps $\tau \gtrsim 10\,\text{s}$.

Based on these results, we conclude that a temporal averaging time of around $T \approx 1\,\text{s}$ marks a transition in representing single-point irradiance variability under mixed-sky conditions. Larger values of $T$ will tend to considerably underestimate variability, while smaller $T$ will increase the complexity of data management and quality control without appreciably improving the representation of variability. Previous studies had judged even higher temporal resolutions between $0.1\,\text{s}$ and $0.4\,\text{s}$ to be adequate, but the respective results were either based on considerably fewer data (Gagné et al., 2016; Torres Lobera et al., 2013), or focused on individual events rather than long-term, climatological variations (Yordanov et al., 2013b). Although there are short-term periods during which an averaging time of $T \approx 1\,\text{s}$ does not capture all changes in irradiance from one second to the next (as illustrated in Fig. 1), such periods occur infrequently based on the records that form the basis of our results. One specific consequence of our results relates to the Baseline Surface Radiation Network (BSRN; McArthur, 2005; Ohmura et al., 1998), which currently records solar irradiance in minute averages. Our results strongly indicate the value of modifying this strategy towards much higher temporal resolutions.

*Data availability.* The data from Alderville and Varennes were kindly provided by Alexandre Gagné at the CanmetENERGY laboratory in Varennes. Samples of the data are available online (Natural Resources Canada, 2016). The raw $0.1\,\text{s}$ irradiance samples from Jülich are courtesy of Andreas Macke and Bomidi Lakshmi Madhavan at the Leibniz Institute for Tropospheric Research (TROPOS) in Leipzig, Germany. A complete set of corresponsing $1\,\text{s}$ averages is available online (Standardized Atmospheric Measurement Data, 2017). The data from Oldenburg were collected by Thomas Schmidt at the Energy Meteorology Group of the University of Oldenburg, and are available from the corresponding author upon reasonable request. The data from Oahu have been made publicly available by the National Renewable Energy Laboratory (NREL) of the United States of America (Sengupta and Andreas, 2010). The data from Tucson have been released online (Lorenzo et al., 2015b) under a Creative Commons Attribution-NonCommercial 4.0 International Public License (CC BY-NC 4.0; Creative Commons, 2013).

*Competing interests.* The authors declare no competing interests.

*Acknowledgements.* We thank Afshin Andreas, Alexander Cronin, Alexandre Gagné, William Holmgren, Antonio Lorenzo, Andreas Macke, Bomidi Lakshmi Madhavan, Thomas Schmidt, and Manajit Sengupta for collecting and sharing their respective irradiance measurements. The time and effort invested in developing and maintaining R (version 3.2.5) by the R Core Team (2016) and the active community of



package authors is gratefully appreciated. All calculations were performed at the HPC Cluster CARL, located at the University of Oldenburg (Germany) and funded by the DFG through its Major Research Instrumentation Programme (INST 184/157-1 FUGG) and the Ministry of Science and Culture (MWK) of the State Lower Saxony. We thank Stefan Harfst at the University of Oldenburg for his support with the cluster. This research was partially funded by the Lower Saxony research network 'Smart Nord', which acknowledges the support of the

5  Lower Saxony Ministry of Science and Culture through the 'Niedersächsisches Vorab' grant program (grant ZN 2764 / ZN 2896). It was also partly funded by the 'Performance Plus' research project through the European Union's Seventh Framework Program for research, technological development and demonstration (grant agreement no. 308991). We also acknowledge funding from the Government of Canada through the ecoENERGY Innovation Initiative (ecoEII) to collect the data in Alderville and Varennes, as well as funding support from the Natural Sciences and Engineering Research Council of Canada.



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





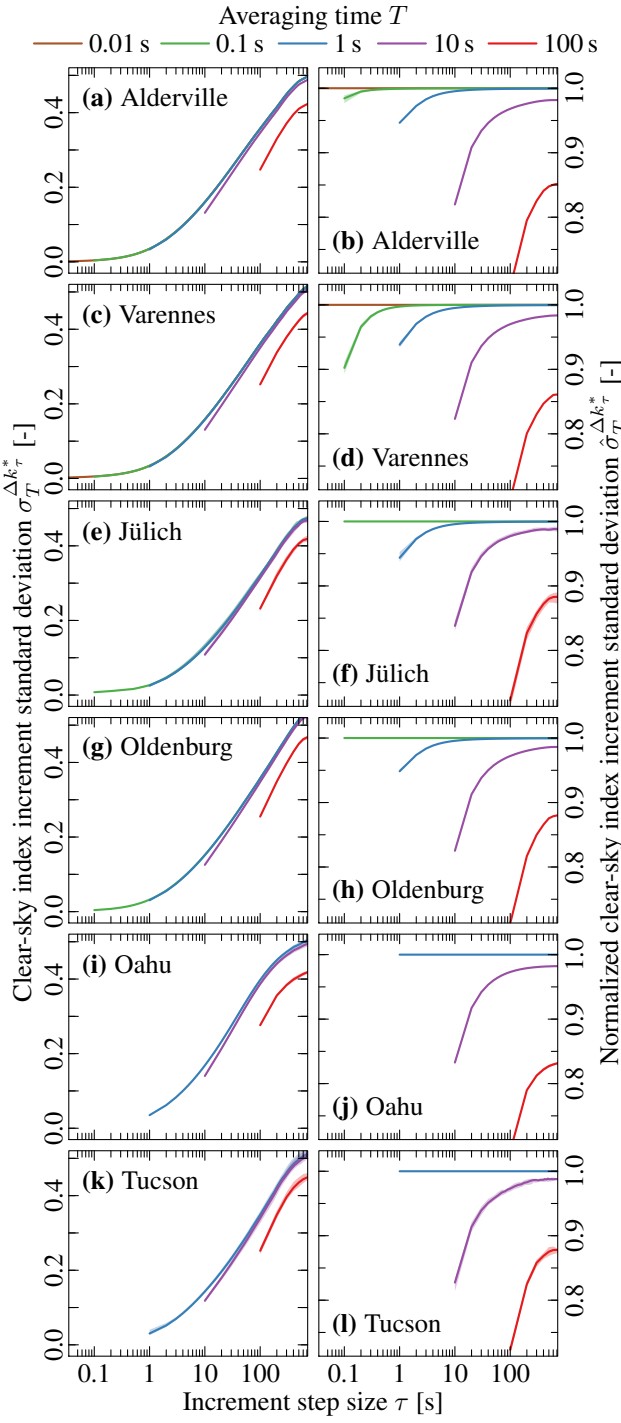

**Figure 6.** Structures of absolute clear-sky index standard deviation $\sigma_T^{\Delta k_\tau^*}$ (left column), and its normalized counterpart $\hat{\sigma}_T^{\Delta k_\tau^*}$ (right column) as functions of increment time $\tau$ for up to five orders of magnitude of averaging time $T = 0.01\,\mathrm{s}$, $T = 0.1\,\mathrm{s}$, $T = 1\,\mathrm{s}$, $T = 10\,\mathrm{s}$, and $T = 100\,\mathrm{s}$ under mixed-sky conditions for all six available locations. The spread across individual single-sensor structures is indicated using semi-transparent coloring, while the solid lines are derived by averaging the single-sensor structures for each $\tau$.





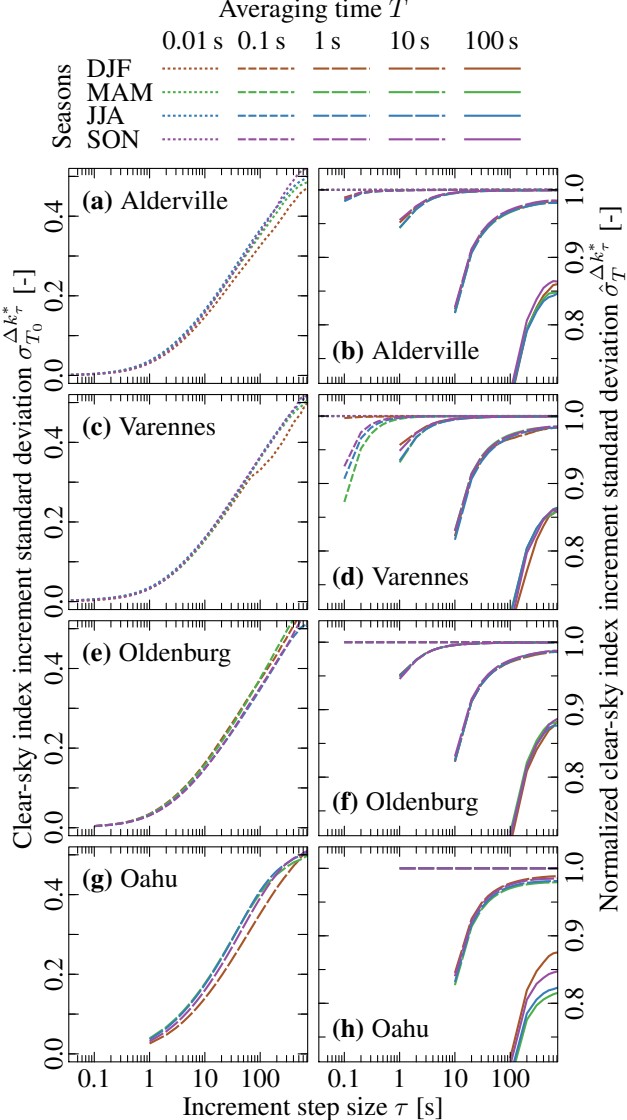

**Figure 7.** As in Fig. 6, for structures separately estimated for winter (December, January, February; DJF), spring (March, April, May; MAM), summer (June, July, August; JJA), and autumn (September, October, November; SON). The spread across single-sensor structures is comparable to that in Fig. 6, but it is not shown here for the sake of facility of inspection.




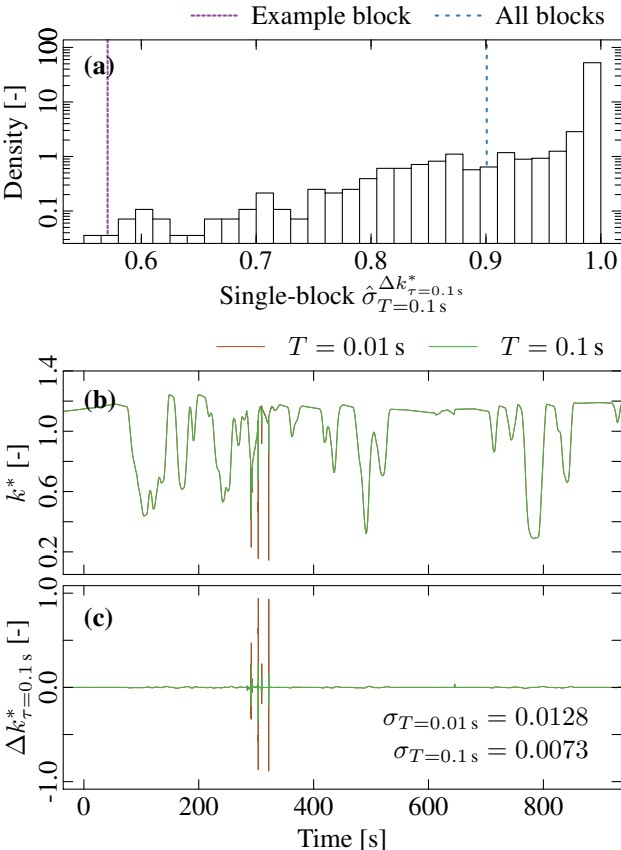

**Figure 8.** (a) The distribution of all Varennes-based single 900-s-block values of normalized clear-sky index increment standard deviation $\hat{\sigma}_T^{\Delta k_\tau^*}$ for $T = 0.1\,\text{s}$ and $\tau = 0.1\,\text{s}$, including an indication of the values corresponding to all blocks (as in Fig. 6) and to an example block featuring a very low $\hat{\sigma}_T^{\Delta k_\tau^*} = 0.57$. Note the logarithmic scaling of the vertical axis. (b) Clear-sky index $k^*$ time series of the 900 s example block for $T = 0.01\,\text{s}$ and $T = 0.1\,\text{s}$, measured in Varennes on 2016-06-23, starting at 07:41:40 EDT. (c) The corresponding clear-sky index increments $\Delta k_\tau^*$ for an increment time step $\tau = 0.1\,\text{s}$, quoting the values of the respective increment standard deviations.