# Peer review of "Effects of temporal averaging on short-term irradiance variability under mixed sky conditions"

_Atmospheric Measurement Techniques, 2017_

## Short Comment (SC1) · 1 Dec 2017

The work presented here has a major interest in the measurements of the downwelling solar irradiance at surface. The authors put emphasis on the production of electricity of PV panels but their work will reach a broader community as solar irradiance has much broader applications. The amount of work is impressive and I may highlight several points. However, I am not a reviewer and I will keep to a few points and questions.

1. The use of clear-sky index

The authors use a clear-sky index which is defined as the ratio of the irradiance G to

that resulting from a model predicting the irradiance in cloud-free conditions Gclear. Such a model is called a clear-sky model.

The authors have used the clear-sky model described in Fontoynont et al. (1988). Actually, this reference does not describe any clear-sky model and should be changed. I guess that the model used is that developed by Dumortier which is a revised version of an original model by Kasten (see Kasten and Young, 1989; Kasten 1996). This model has been criticized and modified to better account for changes in irradiance with ground elevation (see e.g. Geiger et al. 2002).

With respect to the subject of the proposed work, this model has the drawback of using Linke turbidiy factor as input which is unknown at any instant of the time series exploited by the authors. It is likely that the authors use some averaged values of the factor. Hence, Gclear is not the actual cloud-free irradiance but rather a sort of mean value. It follows that variations of k* includes variations in optical properties of the atmospheric composition in cloud-free conditions (aerosols, water vapour mostly). These variables have temporal variabilities that may partly account for the observed variability in k*. As mentioned by the authors, it is likely that variability of k* is mostly due to changing cloudy conditions.

I would not dare to suggest the use of a more advanced clear-sky model such as the McClear model (Lefèvre et al. 2013) that has been validated at several occasions (Lefevre et al. 2013; Eissa et al. 2015; Lefevre et al. 2016). Inputs to McClear are atmospheric properties that are derived from the CAMS chemistry-transport model and the uncertainties of these inputs create uncertainties in Gclear that would be included in the variability of k*.

Nevertheless, I believe that a discussion is missing on the possible role of the selected model and its inputs on the results.

Since the authors are excluding the cases with high solar zenithal angle, I wonder whether the clearness index KT would not better fit. This index is the ratio of G to the

irradiance that is impinging on a horizontal surface at the top of atmosphere G0. G0 is clearly better known that Gclear as it is mostly a function of the sun-earth geometry. This would remove possible ambiguities.

One of these ambiguities is the precise reference to some values of k*. It is clear that changing the clear-sky model will change the values of k*, in terms of means and variance (dynamics). This could at least be mentioned.

2. The cloud-enhanced state

The authors found k* much greater than 1. Though not clearly stated, they call such cases "cloud-enhanced state". I believe that the work will be stronger if the authors show such cases while discussing data itself. The discussion above shows that k* and thus its range (variance, dynamics) depends on the selected clear-sky model and its inputs. Hence, the assumption that "cloud-enhanced state" = "k*»1" must be substantiated.

Note that cloud-enhanced state is also evidenced in KT unambiguously.

3. The method for evidencing the variability

The authors are using appropriate tools for characterizing the variability.

I belive they may use other tools that are better known and that have strong mathematical support that would help for the analysis.

Such a question of variability is not new at all in meteorology. Analysing the change in k* as a function of the time lag (tau) for various time scales (Eq. 4) is a good idea that has been particularly studied in air turbulence by the researchers in the USSR several decades ago (see e.g. Kolmogorov, 1941).

The tool used is called structure function and is an extension of Eq. 4. It has been discovered later in geology where it is called semi-variogram (Matheron 1963). The variogram is the same than the structure function and they differ from the semi-variogram

by a factor 2 (see e.g. Wald, 1989).

The variogram is defined by D(tau) = E[[k*(t+tau)-k*(t)]**2], where E is the mean value.

The difference with Eq. 4 is the square, and the difference with the standard deviation in Fig. 6 is that the variogram do contain the influence of the mean value of [k*(t+tau)-k*(t)] which is removed in the standard deviation.

Using this tool would permit to rely on a considerable literature and a strong mathematical support, with possible links to another considerable literature relating to the Fourier power density spectra, another tool for studying variability with a considerable mathematical background.

4. A few editorial comments.

Page 12, line 6. The deviation for 10 s is not as noticeable in Fig. 6 as claimed. The meaning of "full shadow coverage" is unclear. The "'s" is only for persons in English.

REFERENCES

Kasten F. and Young A.T. (1989) Revised optical air mass tables and approximation formula. Applied Optics, 28 (22), 4735-4738.

Kasten F. (1996) The Linke turbidity factor based on improved values of the integral Rayleigh optical thickness. Solar Energy, 56, 239-244.

Geiger M., Diabaté L., Ménard L., Wald L., 2002. A web service for controlling the quality of measurements of global solar irradiation. Solar Energy, Vol. 73, No 6, pp. 475-480.

Lefèvre, M., A. Oumbe, P. Blanc, B. Espinar, B. Gschwind, Z. Qu, L. Wald, M. Schroedter-Homscheidt, C. Hoyer-Klick, A. Arola, A. Benedetti, J. W. Kaiser, J.-J. Morcrette, 2013. McClear: a new model estimating downwelling solar radiation at ground level in clear-sky conditions. Atmospheric Measurement Techniques, 6, 2403-2418, doi: 10.5194/amt-6-2403-2013.)

Eissa Y., Munawwar S., Oumbe A., Blanc P., Ghedira H., Wald L., Bru H., Goffe D., Validating surface downwelling solar irradiances estimated by the McClear model under cloud-free skies in the United Arab Emirates, Solar Energy, 114, 17-31, 2015, doi:10.1016/j.solener.2015.01.017

Lefèvre M. and Wald L., Validation of the McClear clear-sky model in desert conditions with three stations in Israel. Advances in Science and Research, 13, 21-26, 2016, doi:10.5194/asr-13-21-2016.

Kolmogorov, A.N., 1941. The local structure of turbulence in incompressible viscous fluid for very large Reynolds numbers. Dokl. Akad. Nauk. SSSR, 30, 301-305.

Matheron, G., 1963. Principles of Geostatistics, Economic Geology, 58, 1246-1266.

Wald L., Some examples of the use of structure functions in the analysis of satellite images of the ocean. Photogrammetric Engineering and Remote Sensing, 55, 1487-1490, 1989.
* * *

---

## Referee Comment (RC1) · Anonymous Referee #1 · 12 Dec 2017

General comments:

Lohmann and Monahan present a clear study of irradiance variability as it relates to time averaging. The authors clearly identify the six datasets used in the study, the method used to time-average the data, and the sample standard deviation of clear-sky index and clear-sky index increment as measures of variability. The results and discussion present a clear case for why 1 s averaging may be optimal when variability and data management are a concern. Overall, the manuscript is understandable, organized well, and presents a useful comparison of time-averaged irradiance variability that is applicable across regions. The manuscript could be improved with minor clarifying

points described below.

Specific Comments:

I do not think that Fontoynont et al. 1998 effectively describes the clear-sky model the authors use. Perhaps the more appropriate reference is Dumortier 1996?

Throughout the manuscript, the authors reference possible errors caused by an improper clear-sky model used to generate clear-sky indices. The manuscript would be greatly improved if the authors discussed the specific errors that their chosen clear-sky model introduces. Example time-series of sensor irradiance along with clear-sky model irradiances would be illustrative. The authors will likely find poor fit between the chosen model and some locations. I believe the peak of the distributions above 1.0 shown in Fig. 2 are a direct result of this poor fit, not over-irradiance events. While I do not believe the overall results will change substantially with improved clear-sky modeling, this issue should be addressed in a revised manuscript.

A brief discussion of cloud climatology in Sec. 2, in addition to the example on pg. 8, would be helpful in understanding the datasets. For example, Tucson likely experienced some frontal passages, shallow cumulus, and high cirrus during the study period.

Technical corrections:

Figure 1: The subplot labels (a, b,c, d) are hard to find in the plots. Perhaps moving them outside of the axes is best.

Fig 1 and discussion: after describing how time averaging is performed on 900 s blocks, why is 1000 s used in Fig 1? Perhaps this is only an illustrative example? To avoid confusion I recommend changing from T=1000s to T=900s

---

## Referee Comment (RC2) · O. Perpiñán (Referee) · 9 Mar 2018

O. Perpiñán (Referee)

oscar.perpinan@upm.es

This paper is a evolution of a previous paper by the authors, which I also reviewed. The comments included in this revision are adapted from the comments submitted to that paper.

Once again, this paper is of excellent quality. Data and methods are clearly exposed. Data analysis is exhaustive, and the results are shown with a collection of superb figures. However, there are some issues that could be clarified or improved:

- Both the abstract and the introduction underline the variability of PV power. However,

the analysis is focused on measurements of global irradiation on the horizontal plane (GHI). PV plants produce power with solar irradiance on a inclined plane. It must be noted that, at least on a daily basis, the variability of the effective irradiation incident on inclined planes has been reported to be higher than the variability of irradiation on the horizontal plane: Suri, M., Huld, T., Dunlop, E.D., Albuisson, M., Lefevre, M., Wald, L., 2007. Uncertainties in photovoltaic electricity yield prediction from fluctuation of solar radiation. In: 22nd European Photovoltaic Solar Energy Conference; Perpinan, O., 2009. Statistical analysis of the performance and simulation of a two-axis tracking PV system. Solar Energy 83 (11), 2074–2085.

- In order to remove trends in GHI variability, the authors compute the clear sky index from the GHI measurements. The problem with this index is that the subsequent results are model dependent. In fact, there is not a unique clear sky index because there are several clear sky models to choose. Moreover, most of the models require the use of aerosol measurements or estimations, or assumptions regarding the atmospheric conditions. Therefore, the clear sky model imposes additional uncertainties that were not present in the original data.

- The paper includes a good bibliographic review in the introduction section. However, afterwards the results of the paper are not related to the reviewed papers, and the analysis does not put the results in the context of that review. For example, the bi-modality behaviour (figure 2) and the fdp of increments (figure 5) are strongly related to the results reported in papers included in the bibliographic review. However, there is no comments about it.

- On the other hand, some of the assumptions seem to be arbitrary: for example, the 0.18 threshold in section 2.2 (line 25), and the 95% threshold in section 2.3 (line 5). In my opinion, the arguments provided by the authors should be improved with time series analysis techniques. Therefore, the results reported in the paper cannot be easily compared with others papers.

- Equation 4 uses a simple increment to compute the fluctuations of k*. This approach could be improved as discussed (for example) in Gallego, Cristóbal, Alexandre Costa, Álvaro Cuerva, Lars Landberg, Beatrice Greaves, and Jonathan Collins. 2013. "A Wavelet-Based Approach for Large Wind Power Ramp Characterisation." Wind Energy 16 (2): 257–78. doi:10.1002/we.550.

- The paper will be greatly improved if the authors could publish both the measurements data and the R code, following the recommendations on Reproducible Research: "When publishing computational results, including statistical analyses and simulation, provide links to the source-code (or script) version and the data used to generate the results to the extent that hosting space permits." The Yale Law School Roundtable on Data and Code Sharing. 2010. "Reproducible Research." Computing in Science & Engineering 12 (5). Los Alamitos, CA, USA: IEEE Computer Society: 8–13. doi:10.1109/MCSE.2010.113

---

## Author Comment (AC1) · 28 Apr 2018

**Author Response to Reviews of**

**Effects of temporal averaging on short-term irradiance variability under mixed sky conditions**

Gerald M. Lohmann and Adam H. Monahan

*Atmospheric Measurement Techniques Discussions,* `doi:10.5194/amt-2017-309`
* * *
**RC: Reviewer Comment**,    AR: Author Response,    ☐ Manuscript text

All page and line numbers quoted below refer to the attached latexdiff-version of the manuscript, in which all changes to the original manuscript are highlighted (note that the change-tracking script has not captured changes to the references).

**1. Lucien Wald (Short comment from the scientific community)**

**RC:** **The work presented here has a major interest in the measurements of the downwelling solar irradiance at surface. The authors put emphasis on the production of electricity of PV panels but their work will reach a broader community as solar irradiance has much broader applications. The amount of work is impressive and I may highlight several points. However, I am not a reviewer and I will keep to a few points and questions.**

AR: Thank you very much for commenting on our manuscript. We appreciate your interest in our work along with the constructive feedback and feel that the detailed comments have helped us to improve the quality of the paper.

**1.1. The use of clear-sky index**

**RC:** **The authors use a clear-sky index which is defined as the ratio of the irradiance G to that resulting from a model predicting the irradiance in cloud-free conditions Gclear. Such a model is called a clear-sky model. The authors have used the clear-sky model described in Fontoynont et al. (1988). Actually, this reference does not describe any clear-sky model and should be changed. I guess that the model used is that developed by Dumortier which is a revised version of an original model by Kasten (see Kasten and Young, 1989; Kasten 1996).**

AR: We had made a mistake with the reference and have corrected it in the revised version (cf. page 5, line 27). Thank you for pointing this out.

> Next, we normalize all irradiance measurements $G$ to their respective clear-sky values $G_{clear}$, which we calculate using the clear-sky model described by  Dumortier (1995), to obtain time series of clear-sky index
> $$k^* = \frac{G}{G_{clear}}. \tag{1}$$

**References**

Fontoynont, M., Dumortier, D., Heinemann, D., Hammer, A., Olseth, J., Skarveit, A., Ineichen, P., Reise, C., Page, John, H., Roche, L., Beyer, H.-G., and Wald, L.: SatelLight: A WWW server which provides high quality daylight and solar radiation data for Western and Central Europe, in: 9th Conference on Satellite Meteorology and Oceanography, vol. EUM-P-22, pp. 434–437, American Meteorological Society Ed., Boston, Massachusetts, USA, Paris, France, https://hal-mines-paristech.archives-ouvertes.fr/hal-00472444, 1998.

Dumortier, D.: Modelling global and diffuse horizontal irradiances under cloudless skies with different turbidities, Daylight II, JOU2-CT92- 0144, Final Report, 1995.

RC:   This model has been criticized and modified to better account for changes in irradiance with ground elevation (see e.g. Geiger et al. 2002). With respect to the subject of the proposed work, this model has the drawback of using Linke turbidiy factor as input which is unknown at any instant of the time series exploited by the authors. It is likely that the authors use some averaged values of the factor. Hence, Gclear is not the actual cloud-free irradiance but rather a sort of mean value. It follows that variations of k* includes variations in optical properties of the atmospheric composition in cloud-free conditions (aerosols, water vapour mostly). These variables have temporal variabilities that may partly account for the observed variability in k*. As mentioned by the authors, it is likely that variability of k* is mostly due to changing cloudy conditions. I would not dare to suggest the use of a more advanced clear-sky model such as the McClear model (Lefèvre et al. 2013) that has been validated at several occasions (Lefevre et al. 2013; Eissa et al. 2015; Lefevre et al. 2016). Inputs to McClear are atmospheric properties that are derived from the CAMS chemistry-transport model and the uncertainties of these inputs create uncertainties in Gclear that would be included in the variability of k*. Nevertheless, I believe that a discussion is missing on the possible role of the selected model and its inputs on the results. Since the authors are excluding the cases with high solar zenithal angle, I wonder whether the clearness index KT would not better fit. This index is the ratio of G to the irradiance that is impinging on a horizontal surface at the top of atmosphere G0. G0 is clearly better known that Gclear as it is mostly a function of the sun-earth geometry. This would remove possible ambiguities. One of these ambiguities is the precise reference to some values of k*. It is clear that changing the clear-sky model will change the values of k*, in terms of means and variance (dynamics). This could at least be mentioned.

We have added a corresponding paragraph to the description and discussion of the clear-sky model and its use to derive the clear-sky index (cf. page 6, lines 1–9).

It is important to note that the clear-sky index is not unambiguously defined, because different clear-sky models can yield different values of $G_{clear}$ and thus $k^*$. Equation (1) hence introduces model-dependent uncertainties to the time series that are not present in the original irradiance measurements. In this paper, we use climatological averages of the Linke turbidity factor (Remund et al., 2003) as input to the clear-sky model. $G_{clear}$ thus represents a typical, and not necessarily the effective, clear-sky irradiance on a given day. Consequentially, all locations considered in this study exhibit some periods of suboptimal model performance, with results for Oahu (Tucson) subjectively tending to fit a little better (worse) than for the other locations (not shown). Although variations in $k^*$ may in part be due to variations in atmospheric composition (such as water vapor and aerosols) not accounted for by the clear-sky model, cloud-induced variability is by far the dominant source of irradiance variability on the relatively short timescales considered throughout this paper.

Also, We have added a sentence to the conclusions regarding the McClear model (cf. page 16, lines 29–30).

> Moreover, the modeling of clear-sky irradiance can be improved by using a more advanced clear-sky model, such as the recently validated McClear model (Lefêvre et al., 2013; Eissa et al., 2015; Lefêvre and Wald, 2016).

**1.2. The cloud-enhanced state**

**RC:** **The authors found k\* much greater than 1. Though not clearly stated, they call such cases "cloud-enhanced state". I believe that the work will be stronger if the authors show such cases while discussing data itself. The discussion above shows that k\* and thus its range (variance, dynamics) depends on the selected clear-sky model and its inputs. Hence, the assumption that "cloud-enhanced state" = "k\*»1" must be substantiated. Note that cloud-enhanced state is also evidenced in KT unambiguously.**

 AR: In addition to the new discussion of the clear-sky model mentioned above, we have added a caveat to the text at the point where we discuss cloud-enhancement in the data (cf. page 13, lines 1–5).

> This disparity  is consistent with short-term cloud enhancement: instead of a typical transition going directly from e.g. clear ($k^* \approx 1$) to cloudy ($k^* \approx 0.3$), cloud enhancement immediately precedes (or follows) cloud cover, so typical transitions occur between a cloud-enhanced state ($k^* \gg 1$) and a cloud-covered state ($k^* \approx 0.3$). Note that the unambiguous identification of cloud enhancement is not possible using our values of $k^*$, because of the potential bias and unaccounted for variability included in the clear-sky index we use (cf. discussion in Sec. 2.2).

Also, we have reworded the respective bullet point in the conclusions (cf. page 16, lines 4–9).

> - Clear-sky index increment distributions of 1 s averages and 100 s increment time steps  feature secondary peaks around $\Delta k^* \approx \pm 1$, indicating that transitions of $\pm 1$ in the clear-sky index occur more frequently than changes of $\pm 0.7$. We interpret this result as indicating that transitions going from a cloud-enhanced state ($k^* \gg 1$) to a cloud-covered state ($k^* \approx 0.3$), and vice versa, are more common than transitions directly between clear ($k^* \approx 1$) and cloudy ($k^* \approx 0.3$) states.

**1.3. The method for evidencing the variability**

**RC:** **The authors are using appropriate tools for characterizing the variability. I belive they may use other tools that are better known and that have strong mathematical support that would help for the analysis. Such a question of variability is not new at all in meteorology. Analysing the change in k\* as a function of the time lag (tau) for various time scales (Eq. 4) is a good idea that has been particularly studied in air turbulence by the researchers in the USSR several decades ago (see e.g. Kolmogorov, 1941). The tool used is called structure function and is an extension of Eq. 4. It has been discovered later in geology where it is called semi-variogram (Matheron 1963). The variogram is the same than the structure function and they differ from the semi-variogram by a factor 2 (see e.g. Wald, 1989). The variogram is defined by D(tau) = E[[k\*(t+tau)-k\*(t)]\*\*2], where E is the mean value. The difference with Eq. 4 is the square, and the difference with the standard deviation in Fig. 6 is that the variogram do contain the influence of the mean value of**

**[k\*(t+tau)-k\*(t)] which is removed in the standard deviation. Using this tool would permit to rely on a considerable literature and a strong mathematical support, with possible links to another considerable literature relating to the Fourier power density spectra, another tool for studying variability with a considerable mathematical background.**

AR: Thank you for suggesting these other tools. We have added a corresponding paragraph to the conclusions (cf. page 16, lines 30–32).

> Also, it may be beneficial to extend the simple increment procedure (cf. Eq. 4) by considering the semi-variogram (Matheron, 1963) and variogram (Wald, 1989), or by using a wavelet-based approach such as that discussed by Gallego et al. (2013).

**1.4. A few editorial comments**

RC: **Page 12, line 6. The deviation for 10 s is not as noticeable in Fig. 6 as claimed.**

AR: Agreed. We have slightly reworded the sentence in question in order to better distinguish between the characteristics of the 10 s and 100 s structures (cf. page 13, line 9).

> The structures of $\sigma_T^{\Delta k_\tau^*}$ (left column) are effectively indistinguishable for $T = 0.01$ s, $T = 0.1$ s, and $T = 1$ s, while structures associated with longer averaging periods of $T = 10$ s  ($T = 100$ s ) deviate slightly (notably) from their high-resolution counterparts.

RC: **The meaning of "full shadow coverage" is unclear.**

AR: We have slightly reworded the sentences in question (cf. page 9, lines 5–6; page 12, lines 12–13).

> The two modes are located slightly above $k^* \simeq 1$ (corresponding to times of clear skies), and within a range of $0.3 \lesssim k^* \lesssim 0.5$ (corresponding to times of cloud shadow coverage), with modest differences between locations.

> Comparing the distributions of clear-sky index for $T = 1$ s (Fig. 2) with those of clear-sky index increments for $T = 1$ s and $\tau = 100$ s (Fig. 5) allows further characterization of transitions from  times of clear skies to times of cloud shadow coverage (and vice versa).

RC: **The "'s" is only for persons in English.**

AR: We disagree about this nuance and prefer not to change our text in this regard.

[revised manuscript text omitted]

---

## Author Comment (AC2) · 28 Apr 2018

**Author Response to Reviews of**

**Effects of temporal averaging on short-term irradiance variability under mixed sky conditions**

Gerald M. Lohmann and Adam H. Monahan

*Atmospheric Measurement Techniques Discussions,* `doi:10.5194/amt-2017-309`
* * *
**RC: Reviewer Comment**,    AR: Author Response,    ☐ Manuscript text

All page and line numbers quoted below refer to the attached latexdiff-version of the manuscript, in which all changes to the original manuscript are highlighted (note that the change-tracking script has not captured changes to the references).

**1. Anonymous Referee #1**

**1.1. General comments**

**RC:** **Lohmann and Monahan present a clear study of irradiance variability as it relates to time averaging. The authors clearly identify the six datasets used in the study, the method used to time-average the data, and the sample standard deviation of clear-sky index and clear-sky index increment as measures of variability. The results and discussion present a clear case for why 1 s averaging may be optimal when variability and data management are a concern. Overall, the manuscript is understandable, organized well, and presents a useful comparison of time-averaged irradiance variability that is applicable across regions. The manuscript could be improved with minor clarifying points described below.**

**AR:** Thank you very much for reviewing our manuscript. We appreciate the positive feedback and feel that the detailed comments have helped us to improve the quality of the paper.

**1.2. Specific comments**

**RC:** **I do not think that Fontoynont et al. 1998 effectively describes the clear-sky model the authors use. Perhaps the more appropriate reference is Dumortier 1996?**

**AR:** We had made a mistake with the reference and have corrected it in the revised version (cf. page 5, line 27). Thank you for pointing this out.

> Next, we normalize all irradiance measurements $G$ to their respective clear-sky values $G_{clear}$, which we calculate using the clear-sky model described by Dumortier (1995), to obtain time series of clear-sky index
>
> $$k^* = \frac{G}{G_{clear}}. \tag{1}$$

**References**

Fontoynont, M., Dumortier, D., Heinemann, D., Hammer, A., Olseth, J., Skarveit, A., Ineichen, P., Reise, C., Page, John, H., Roche, L., Beyer, H.-G., and Wald, L.: SatelLight: A WWW server which provides high quality daylight and solar radiation data for Western and Central Europe, in: 9th Conference on Satellite Meteorology and Oceanography, vol. EUM-P-22, pp. 434–437, American Meteorological Society Ed., Boston, Massachusetts, USA, Paris, France, https://hal-mines-paristech.archives-ouvertes.fr/hal-00472444, 1998.

Dumortier, D.: Modelling global and diffuse horizontal irradiances under cloudless skies with different turbidities, Daylight II, JOU2-CT92- 0144, Final Report, 1995.

**RC:** Throughout the manuscript, the authors reference possible errors caused by an improper clear-sky model used to generate clear-sky indices. The manuscript would be greatly improved if the authors discussed the specific errors that their chosen clear-sky model introduces. Example time-series of sensor irradiance along with clear-sky model irradiances would be illustrative. The authors will likely find poor fit between the chosen model and some locations. I believe the peak of the distributions above 1.0 shown in Fig. 2 are a direct result of this poor fit, not over-irradiance events. While I do not believe the overall results will change substantially with improved clear-sky modeling, this issue should be addressed in a revised manuscript.

**AR:** All locations considered in this study exhibit some periods of suboptimal model performance, with results for Oahu (Tucson) subjectively tending to fit a little better (worse) than for the other locations. A thorough quantification of the model performance at the different locations goes beyond the scope of our paper. Please have a look at the following plots, showing representative examples of relatively bad model fits (left column) and relatively good model fits (right column) for all locations (and en passant also including some examples of over-irradiance):

[Figure]

[Figure]

Note that we did not limit ourselves to complete clear-sky days in these plots, because we did not analyze any periods of complete clear-sky conditions in the paper. We also found different clear-sky model performances for different sensors at the same locations and during the same time, presumably due to small variations in the sensors' orientations and calibration (not shown). Hence, the possible errors included in the clear-sky index can originate both from the clear-sky model and from imperfect sensor measurements. In any case, these errors only affect the absolute values of the clear-sky index but not the shapes of corresponding (increment) distributions, and certainly not how these distributions' shapes change with averaging (because of the relatively slow temporal rate of change in the clear sky model).

We have added a corresponding paragraph to the description and discussion of the clear-sky model and its use to derive the clear-sky index (cf. page 6, lines 1–9).

> It is important to note that the clear-sky index is not unambiguously defined, because different clear-sky models can yield different values of $G_{clear}$ and thus $k^*$. Equation (1) hence introduces model-dependent uncertainties to the time series that are not present in the original irradiance measurements. In this paper, we use climatological averages of the Linke turbidity factor (Remund et al., 2003) as input to the clear-sky model. $G_{clear}$ thus represents a typical, and not necessarily the effective, clear-sky irradiance on a given day. Consequentially, all locations considered in this study exhibit some periods of suboptimal model performance, with results for Oahu (Tucson) subjectively tending to fit a little better (worse) than for the other locations (not shown). Although variations in $k^*$ may in part be due to variations in atmospheric composition (such as water vapor and aerosols) not accounted for by the clear-sky model, cloud-induced variability is by far the dominant source of irradiance variability on the relatively short timescales considered throughout this paper.

**RC:** **A brief discussion of cloud climatology in Sec. 2, in addition to the example on pg. 8, would be helpful in understanding the datasets. For example, Tucson likely experienced some frontal passages, shallow cumulus, and high cirrus during the study period.**

AR: We have added a corresponding short paragraph and table providing a basic overview of season- and cloud-elevation-specific cloud climatology by quoting the respective average cloud amount during the day from the Climatic Atlas of Clouds Over Land and Ocean (Eastman et al., 2014) (cf. page 5, lines 13–18 and Table 3 top area of page 6)

> For each location, Table 3 provides a basic overview of season- and cloud-elevation-specific cloud climatology based on the respective average cloud amount during the day from the Climatic Atlas of Clouds Over Land and Ocean (Eastman et al., 2014). The four mid-latitude locations feature comparable values of cloud amount across seasons and altitude ranges, while Oahu and Tucson stand out with appreciably smaller values for low level (Tucson only) and middle level clouds (both). Although the two locations also systematically exhibit smaller values than the others for high level cloud amount, these differences are not as pronounced.

**Table 3.** Average cloud amount [%] during the day for each season (winter: DJF, spring: MAM, summer: JJA, autumn: SON) extracted from the Climatic Atlas of Clouds Over Land and Ocean (Eastman et al., 2014) for all six locations in question. Values are grouped by cloud altitude, with the low level including fog, stratus (St), stratocumulus (Sc), cumulus (Cu), and cumulonimbus (Cb); the middle level consisting of nimbostratus (Ns), altostratus (As), and altocumulus (Ac); and the high level representing all cirriform clouds combined (Hahn and Warren, 2007). For the short-term measurement campaigns near Jülich and Tucson, only those data corresponding to the study periods are quoted. Note that Jülich and Oldenburg are located within the same grid box of the cloud atlas and hence feature identical values.

| Location | All low level clouds | | | | All middle level clouds | | | | All high level clouds | | | |
|---|---|---|---|---|---|---|---|---|---|---|---|---|
| | DJF | MAM | JJA | SON | DJF | MAM | JJA | SON | DJF | MAM | JJA | SON |
| Alderville | 55 | 42 | 35 | 49 | 37 | 32 | 25 | 31 | 26 | 29 | 26 | 25 |
| Varennes | 42 | 41 | 40 | 53 | 43 | 36 | 32 | 40 | 23 | 27 | 24 | 23 |
| Jülich | – | 49 | 47 | – | – | 28 | 27 | – | – | 23 | 20 | – |
| Oldenburg | 59 | 49 | 47 | 52 | 32 | 28 | 27 | 30 | 24 | 23 | 20 | 23 |
| Oahu | 45 | 48 | 45 | 43 | 12 | 11 | 8 | 10 | 11 | 16 | 14 | 17 |
| Tucson | – | 7 | 8 | – | – | 8 | 13 | – | – | 20 | 13 | – |

**1.3. Technical corrections**

**RC: Figure 1: The subplot labels (a, b,c, d) are hard to find in the plots. Perhaps moving them outside of the axes is best.**

AR: In order to improve legibility of the labels, we have adjusted their positioning and slightly changed the scaling of the vertical axes in all panels of Fig. 1 (cf. top area on page 8).

[Figure]

**RC:** **Fig 1 and discussion: after describing how time averaging is performed on 900 s blocks, why is 1000 s used in Fig 1? Perhaps this is only an illustrative example? To avoid confusion I recommend changing from T=1000s to T=900s**

**AR:** We would prefer to keep the averaging time scale $T = 1000\,\mathrm{s}$ in Fig. 1 for illustrative purposes in order to still provide the reader with an intuitive range of 6 orders of magnitude in the legend. Throughout the manuscript text, these $1000\,\mathrm{s}$ averages from Fig. 1 are neither mentioned nor used. In order to avoid confusion we have thus adapted the figure caption of Fig. 1 to make ourselves more clear (cf. caption of Fig. 1 in top area on page 8).
* * *

[revised manuscript text omitted]

---

## Author Comment (AC3)

**Author Response to Reviews of**

**Effects of temporal averaging on short-term irradiance variability under mixed sky conditions**

Gerald M. Lohmann and Adam H. Monahan

*Atmospheric Measurement Techniques Discussions,* `doi:10.5194/amt-2017-309`
* * *
**RC: Reviewer Comment**,     AR: Author Response,     ☐ Manuscript text

All page and line numbers quoted below refer to the attached latexdiff-version of the manuscript, in which all changes to the original manuscript are highlighted (note that the change-tracking script has not captured changes to the references).

**1. Oscar Perpiñán (Referee #2)**

RC: **This paper is a evolution of a previous paper by the authors, which I also reviewed. The comments included in this revision are adapted from the comments submitted to that paper. Once again, this paper is of excellent quality. Data and methods are clearly exposed. Data analysis is exhaustive, and the results are shown with a collection of superb figures. However, there are some issues that could be clarified or improved:**

AR: Thank you very much for taking the time to review another one of our manuscripts! We appreciate the positive feedback and feel that the detailed comments have – yet again – helped us to improve the quality of the paper.

RC: **Both the abstract and the introduction underline the variability of PV power. However, the analysis is focused on measurements of global irradiation on the horizontal plane (GHI). PV plants produce power with solar irradiance on a inclined plane. It must be noted that, at least on a daily basis, the variability of the effective irradiation incident on inclined planes has been reported to be higher than the variability of irradiation on the horizontal plane: Suri, M., Huld, T., Dunlop, E.D., Albuisson, M., Lefevre, M., Wald, L., 2007. Uncertainties in photovoltaic electricity yield prediction from fluctuation of solar radiation. In: 22nd European Photovoltaic Solar Energy Conference; Perpinan, O., 2009. Statistical analysis of the performance and simulation of a two-axis tracking PV system. Solar Energy 83 (11), 2074–2085.**

AR: We agree and have included corresponding statements in the introduction (cf. page 2, lines 1–4) and conclusions (cf. page 16, lines 26 ff.).

> While early studies focused on hourly to daily timescales (e.g. Liu and Jordan, 1960), later analyses were often geared towards finer temporal resolutions between 300 s and 0.01 s (Table 1). The irradiance data considered in such studies were typically collected on the horizontal plane, although PV systems commonly feature tilted modules. With regards to PV power applications,  it is thus important to note that variability in irradiance on an inclined plane has been shown to be higher than on the horizontal, at least on a daily basis (Perpiñán, 2009; Suri et al., 2007). (...) There is no consensus as to the proper temporal resolution of irradiance measurements needed to capture all relevant variability.

> To carry the present research questions further, analyzing high-resolution irradiance measurements in the plane-of-array (POA) of existing PV systems (instead of focussing exclusively on global horizontal irradiance) would lead to results with higher direct applicability to PV power variability (although contrasting different sites will not be straight forward for POA irradiance).

**RC:** **In order to remove trends in GHI variability, the authors compute the clear sky index from the GHI measurements. The problem with this index is that the subsequent results are model dependent. In fact, there is not a unique clear sky index because there are several clear sky models to choose. Moreover, most of the models require the use of aerosol measurements or estimations, or assumptions regarding the atmospheric conditions. Therefore, the clear sky model imposes additional uncertainties that were not present in the original data.**

AR: We have added a corresponding paragraph to the description and discussion of the clear-sky model and its use to derive the clear-sky index (cf. page 6, lines 1–9).

> It is important to note that the clear-sky index is not unambiguously defined, because different clear-sky models can yield different values of $G_{clear}$ and thus $k^*$. Equation (1) hence introduces model-dependent uncertainties to the time series that are not present in the original irradiance measurements. In this paper, we use climatological averages of the Linke turbidity factor (Remund et al., 2003) as input to the clear-sky model. $G_{clear}$ thus represents a typical, and not necessarily the effective, clear-sky irradiance on a given day. Consequentially, all locations considered in this study exhibit some periods of suboptimal model performance, with results for Oahu (Tucson) subjectively tending to fit a little better (worse) than for the other locations (not shown). Although variations in $k^*$ may in part be due to variations in atmospheric composition (such as water vapor and aerosols) not accounted for by the clear-sky model, cloud-induced variability is by far the dominant source of irradiance variability on the relatively short timescales considered throughout this paper.

**RC:** **The paper includes a good bibliographic review in the introduction section. However, afterwards the results of the paper are not related to the reviewed papers, and the analysis does not put the results in the context of that review. For example, the bimodality behaviour (figure 2) and the fdp of increments (figure 5) are strongly related to the results reported in papers included in the bibliographic review. However, there is no comments about it.**

AR: We have added corresponding comments to the discussions of Figs 2 (cf. page 9, lines 8–10) and 5 (cf. page 11, lines 9–12), respectively.

>  The distinct bimodality of the distributions of 1 s averages is consistent with previous findings of e.g., Jurado et al. (1995), Lohmann et al. (2016), Munkhammar et al. (2017), Schmidt et al. (2016), Skartveit and Olseth (1992), Suehrcke and McCormick (1988), and Woyte et al. (2007). This feature is substantially weaker for the distributions of 100 s averages.

> The finding of such non-Gaussian increment distributions is consistent with e.g., Hinkelman (2013) and Perpiñán and Lorenzo (2011). The systematic changes in the distributions' shapes for different averaging intervals and increment time steps qualitatively agree with reports published by e.g., Anvari et al. (2016), van Haaren et al. (2014), Lave et al. (2012), and Marcos et al. (2011).

**RC:** **On the other hand, some of the assumptions seem to be arbitrary: for example, the 0.18 threshold in section 2.2 (line 25), and the 95% threshold in section 2.3 (line 5). In my opinion, the arguments provided by the authors should be improved with time series analysis techniques. Therefore, the results reported in the paper cannot be easily compared with others papers.**

 AR: These values were determined subjectively based on inspection of the data. Small changes to the values do not substantially change the results. We have adapted the two corresponding paragraphs (cf. page 6, lines 10–14 and page 7 lines 6–8) to make ourselves more clear.

> In order to extract mixed sky periods from the clear-sky index time series, we employ a  subjective classification scheme based on dividing all time series into non-overlapping 900 s blocks, and calculating the individual blocks' clear-sky index standard deviations. If the standard deviation of $k^*$ exceeds a fixed threshold (taken to be 0.18, based on the results of Lohmann et al., 2016), we classify the corresponding 900 s block as mixed, and retain it for our analyses. Small changes to the value of this threshold do not substantially change the results.

> If fewer than 95% of the highest-resolution data within a period $T$ are available, the resulting average is classified as missing. Otherwise the missing values in the raw data are simply neglected in the computation. We found this subjective threshold of 95% to be a good compromise  between minimizing the number of missing data  and maintaining the robustness of the mean estimates. Slight changes of the threshold value do not noticably affect the results. With appreciably higher thresholds, however, a small number of missing high-resolution data points could cause unreasonably many long-term averages to be missing; while for considerably lower thresholds, there would be too much variation in the number of data points entering the computation of means of nominal averaging time $T$.

**RC:** **Equation 4 uses a simple increment to compute the fluctuations of k\*. This approach could be improved as discussed (for example) in Gallego, Cristóbal, Alexandre Costa, Álvaro Cuerva, Lars Landberg, Beatrice Greaves, and Jonathan Collins. 2013. "A Wavelet-Based Approach for Large Wind Power Ramp Characterisation." Wind Energy 16 (2): 257–78. doi:10.1002/we.550.**

 AR: We have included a corresponding statement in the conclusions (cf. page 16, line 32).

> Also, it may be beneficial to extend the simple increment procedure (cf. Eq. 4) by considering the semi-variogram (Matheron, 1963) and variogram (Wald, 1989), or by using a wavelet-based approach such as that discussed by Gallego et al. (2013).

**RC:** **The paper will be greatly improved if the authors could publish both the measurements data and the R code, following the recommendations on Reproducible Research: "When publishing computational results, including statistical analyses and simulation, provide links to the source-code (or script) version and the data used to generate the results to the extent that hosting space permits."**

**The Yale Law School Roundtable on Data and Code Sharing. 2010. "Reproducible Research." Computing in Science & Engineering 12 (5). Los Alamitos, CA, USA: IEEE Computer Society: 8–13. doi:10.1109/MCSE.2010.113**

AR: The data availability has been documented in the corresponding subsection of the manuscript (cf. page 17, lines 1–7). Unfortunately, we are not at liberty to openly (re-)publish the data supplied by our partners. However, we agree that it was a drawback of the original manuscript that the Oldenburg data were not publicly available at all. We have now initiated their publication under a Creative Commons Licence Schmidt and Lohmann (2018). Moreover, we will publish the clear-sky model output as well as the computer code used in the analyses as a supplement to the final version of the paper, once it is accepted for publication by AMT.

> *Data availability.* The data from Alderville and Varennes were kindly provided by Alexandre Gagné at the CanmetENERGY laboratory in Varennes. Samples of the data are available online (Natural Resources Canada, 2016). The raw 0.1 s irradiance samples from Jülich are courtesy of Andreas Macke and Bomidi Lakshmi Madhavan at the Leibniz Institute for Tropospheric Research (TROPOS) in Leipzig, Germany. A complete set of  corresponding 1 s averages is available online (Standardized Atmospheric Measurement Data, 2017). The data from  Oahu have been made publicly available by the National Renewable Energy Laboratory (NREL) of the United States of America (Sengupta and Andreas, 2010). The data from  Oldenburg (Schmidt and Lohmann, 2018) and Tucson (Lorenzo et al., 2015b) have been released online under a Creative Commons Attribution-NonCommercial 4.0 International Public License (CC BY-NC 4.0; Creative Commons, 2013).

[revised manuscript text omitted]